# Selection of Leptin Surrogates by a General Phenotypic Screening Method for Receptor Agonists

**DOI:** 10.3390/biom14040457

**Published:** 2024-04-09

**Authors:** Tao Wang, Xixi Chen, Guang Yang, Xiaojie Shi

**Affiliations:** 1Shanghai Institute for Advanced Immunochemical Studies, ShanghaiTech University, Shanghai 201210, China; wangtao1@shanghaitech.edu.cn (T.W.); chenxx2@shanghaitech.edu.cn (X.C.); yangguang@shanghaitech.edu.cn (G.Y.); 2School of Life Science and Technology, ShanghaiTech University, Shanghai 201210, China

**Keywords:** phenotypic screening, high-throughput screening, cytokine surrogate, receptor agonist, biological receptor activation-dependent cell survival (BRADS)

## Abstract

There is a high demand for agonist biomolecules such as cytokine surrogates in both biological and medicinal research fields. These are typically sourced through natural ligand engineering or affinity-based screening, followed by individual functional validation. However, efficient screening methods for identifying rare hits within immense libraries are very limited. In this research article, we introduce a phenotypic screening method utilizing biological receptor activation-dependent cell survival (BRADS). This method offers a high-throughput, low-background, and cost-effective approach that can be implemented in virtually any biochemical laboratory setting. As a proof-of-concept, we successfully identified a surrogate for human leptin following a two-week cell culture process, without the need for specialized high-throughput equipment or reagents. This surrogate effectively emulates the activity of native human leptin in cell validation assays. Our findings not only underscore the effectiveness of BRADS but also suggest its potential applicability to a broad range of biological receptors, including Notch and GPCRs.

## 1. Introduction

The inception of combinatorial library technology has significantly impacted drug discovery [1,2], most notably in the development of Adalimumab [3]. Central to combinatorial libraries is the principle of infinite diversity, enabling the screening of vast molecular pools against virtually any target in vitro. This mimics the functionality of the natural immune system in vivo, without the drawbacks of long-term immune responses or tolerance issues [4,5]. By utilizing this synthetic immune system framework, numerous high-quality therapeutic antibodies targeting a wide range of drug targets have been identified by us. These targets include human proteins such as GPCRs, ion channels, gap junction hemichannels, tumor-associated antigens, and exogenous proteins from pathogens like the SARS-CoV-2 spike protein [6,7,8].

Affinity-based high-throughput screening ensures the identification of rare members in the library with immense diversity. Nonetheless, the direct selection of cytokine surrogates as agonists for biological receptors, which are crucial targets in pharmaceuticals, poses a significant challenge. To address this, various phenotypic screening systems have been developed, such as the Tango system for GPCRs and JAK/STAT reporters for cytokine receptors [9,10]. These systems typically rely on reporter genes and high-throughput platforms such as automatic microwell plate systems or flow cytometry. For the antibody library, an autocrine cell-based selection method was developed. In this system, each antibody in the library is tethered to the surface of an individual cell, influencing that same cell [10,11,12]. Cells exhibiting the desired phenotype are thus sortable, linking the phenotype directly to the antibody’s genotype, which can then be identified through DNA sequencing. This approach has facilitated the discovery of numerous functional antibodies, particularly those capable of modulating cell fate [13,14]. Despite its advantages, this method faces challenges such as the non-specific background signal in reporter systems and the requirement for complex screening systems, leading to increased costs and potentially high false-positive rates, necessitating further validation assays.

In this paper, we present a novel phenotypic screening method for identifying agonist biomolecules, termed the biological receptor activation-dependent survival (BRADS) of cells. This approach selects and engineers a factor-dependent cell strain that proliferates in response to the activation of a specific cytokine receptor of interest, while other cells are eliminated in the absence of the initial growth factor. Following a two-week cell culture after withdrawal of the growth factor, cells harboring an agonist antibody in the autocrine format are naturally enriched and can be easily decoded by sequencing. As a proof-of-concept, we applied this method to select an agonist antibody for the human leptin receptor, a key regulator in body fat metabolism and various physiological processes belonging to the type I cytokine receptor. Stimulation by its natural ligand leptin induces a decrease in food intake and an increase in energy consumption, mainly by inducing anorexigenic factors and suppressing orexigenic neuropeptides [15,16,17]. This pathway also regulates bone mass [18], reproductive function [19], insulin resistance [20], and lymphopoiesis [21]. Leptin surrogates, which have much longer half-life in the body, have potential in the treatment of obesity and diabetes [22,23]. Our approach successfully identified agonists from an antibody pool with a diversity of at least 10^6^ in just two weeks, without the need for sorting machines or automatic high-throughput equipment. This agonist was further validated as a leptin surrogate in in vitro cell functions, demonstrating its potential for the treatment of obesity and diabetes.

## 2. Materials and Methods

### 2.1. Cell Culture

The murine cell line Ba/F3 (murine pro B cell line, DSMZ #ACC-300, Brunswick, Germany) and its corresponding BRADS strains were maintained in RPMI-1640 (#SH30027.FS, Hyclone, Logan, UT, USA) containing 10% (*v*/*v*) FBS (#100-500, Gemini, New York, NY, USA) and 2 ng/mL murine IL-3 (#CP39A, Novoprotein, Suzhou, China). EML (murine multipotent hematopoietic cell line, ATCC #CRL-3610, kindly gifted by Yu’s Lab in Tongji Univ.) was maintained in RPMI-1640 containing 20% (*v*/*v*) FBS and 200 ng/mL murine SCF (#CP39A, Novoprotein, Suzhou, China). 293T-*h*LeptR stable cell lines with or without SIE-GFP / SIE-Luciferase reporter systems were maintained in DMEM (#SH30243.01, Hyclone, Logan, UT, USA) containing 10% (*v*/*v*) FBS. The conditions in the screening process are described below. All cells were cultured at 37 °C with 5% CO_2_.

### 2.2. Library Screening

First, a human combinatorial single-chain fragment variable (scFv) library was screened against the protein antigen of the ECD of the human leptin receptor. Then, the resulting focused sub-library (with a diversity around 10^6^) was constructed into an autocrine scFv library in lentiviral vector form. Meanwhile, the Ba/F3 cell line was transduced by the lentivirus containing the human leptin receptor gene (coding DNA sequence). Single clones of *h*LepR-expressing stable cells were selected, and one of them was used for library screening. Afterwards, 5 × 10^7^ of the stable cells were transduced with the concentrated lentivirus carrying the *h*LepR-focused autocrine library to reach near 60% positivity by centrifugation at 800× *g* for 70 min in the presence of 10 μg/mL polybrene for each screening batch. Twenty-four hours after transduction, the growth factor was derived from the cells by introducing a fresh medium. The cells were resuspended at 1 × 10^6^ cells/mL and cultured for 2 weeks. Cells without transduction were set as the negative control. Cells were collected when the viability of the control cells became nearly zero. Cell viability was determined by live cell counting using Trypan blue (#T8154, Sigma, St. Louis, MO, USA). ScFv sequences were amplified by PCR with the primer pair flanking scFv and were cloned and transformed into competent cells. Eighty singe colonies were picked and sequenced by Sanger sequencing. Sequences were aligned using MEGA software (11.0.13), germline analysis was carried out online using IgBLAST, and Weblogo 3.7 was used for the visualization of the CDRH3 alignment.

scFv sequences of the hit clones were cloned into a pFuse vector to make the scFv-hFc fusion form. Then, the antibodies were transiently expressed in HEK-293F suspension cells (#R79007, Gibco, Grand Island, NY, USA) cultured in FreeStyle 293 medium (#12338026, Gibco), followed by purification using Mabselect columns (#17-5199-01, Cytiva, Shanghai, China). Soluble form of this antibody was used for the subsequent validation.

### 2.3. Binding of the Antibody to hLepR on Live Cells

Flow cytometry was used to determine the binding affinity of the selected antibody to the *h*LepR expressed on the surface of live cells. The stable *h*LepR-expressing 293T cells (in house) were detached using accutase (#07920, STEMCELL, Vancouver, Canada). Then, the cells were washed with PBS twice and incubated with the antibody at different concentrations, from 33.4 pM to 6.7 nM, at a density of 1 × 10^6^ cells/100 μL cell staining buffer (#420201, Biolegend, San Diego, CA, USA) per sample on ice for 20 min. The resulting cells were subsequently stained with anti-hIgG-Alexa Fluor 555 (#A21433, Invitrogen, Carlsbad, CA, USA) on ice for 20 min, followed by washing with PBS. Finally, the cells were resuspended in the cell staining buffer and detected by flow cytometry (CytoFLEX, Beckman Coulter, Brea, CA, USA).

### 2.4. Western Blot for STAT Phosphorylation

LepR-Ba/F3 cells were incubated in serum-free medium overnight and then treated with 100 ng/mL leptin (#Z02962-1; GenScript, Nanjing, China) or 500 ng/mL agonist-type antibody at 37 °C, 5% CO_2_ for 30 min. The cells were collected and washed once with ice-cold PBS containing a Halt protease/phosphatase inhibitor cocktail (#78446; Thermo Scientific, Waltham, MA, USA), followed by 30 min of lysis with RIPA buffer containing a protease/phosphatase inhibitor cocktail on ice with shaking. The resultant supernatant was collected by centrifuging at 14,000× *g* rpm for 10 min. The amount of total STAT3 or STAT5 and phosphorylated STAT3 or STAT5 were determined by a western blot analysis of the cell lysates’ supernatant using anti-STAT3 antibody (#32500; Abcam, Cambridge, UK), anti-STAT5 antibody (#9363T; Cell Signaling Technology), anti-pSTAT3 (Tyr705) (#76315; Abcam), and anti-pSTAT5 (#4322P; Cell Signaling Technology, Danvers, MA, USA), respectively.

### 2.5. Agonist Activity Determination by SIE-Luciferase/GFP Reporter System

The SIE-luciferase reporter cell strain (based on HEK 293T) with exogenous *h*LepR expression was plated to white opaque 96-well plates at a density of 4 × 10^5^ cells/mL (50 μL per well). The antibody was 1:3 serially diluted in the complete medium to from 200 nM to 40 fM. Diluted antibodies with different concentrations were added to the cells (50 μL per well) and incubated at 37 °C for 18 h. Three replicates were set for each concentration. Luciferase activity representing the agonist activity was detected with the FIREFLYGLO Luciferase Reporter Assay kit (#MA0519-2; Meilunbio, Dalian, China) according to the manufacturer’s instruction. Briefly, 40 μL of FIREFLYGLO Luciferase substrate solution was added to each well, and then the luminescence intensity of each well was measured on a microplate reader (Enspire, PerkinElmer, Waltham, MA, USA).

The SIE-GFP reporter cell strain (based on HEK 293T) with exogenous *h*LepR expression was seeded on 6-well plates at 4 × 10^5^ cells/mL (2 mL per well). Human leptin and sB12 were added to the corresponding wells at a final concentration of 500 ng/mL. The cells were cultured overnight, and cellular GFP expression was detected by fluorescence microscopy and flow cytometry.

### 2.6. Factor-Dependent Cell Proliferation Assay

BRADS cells were washed 3 times with PBS to remove medium residues and plated in 96-well microwell plates at a density of 10,000 cells in 50 μL per well. Agonists (human leptin or sB12) were 1:3 serially diluted from 100 nM to 0.3 pM and mixed with the cells (50 μL per well). The cells were cultured at 37 °C, 5% CO_2_ for an additional 72 h. Cell Counting Kit-8 Cell Proliferation Assay (#MA0218; Meilunbio) was used according to the manufacturer’s instructions. Briefly, 10 μL of the reaction solution was added to each well, mixed, and incubated at 37 °C for 2 h. The absorbance at 490 nm of each well was detected on a microplate reader (Enspire, PerkinElmer).

### 2.7. Statistical Analyses

Plot data are expressed as means ± standard deviation (S.D.) unless otherwise indicated. Plot data were analyzed with GraphPad (Prism 9.0). Four-parameter non-linear regression was used for curve fitting and IC_50_ calculation. A two-tailed Student’s *t*-test or one-way ANOVA (two groups or more than two groups) was used for group comparisons (data sets normally distributed with equal variances; significance is indicated as *p* value in the figure legends).

## 3. Results

### 3.1. Cells for BRADS-Based Screening

To construct a screening system based on cell survival (Figure 1A, upper), we initially compared the IL-3-dependent pro-B murine cell line Ba/F3 with the SCF-dependent multipotent hematopoietic murine cell line EML in terms of factor-dependent growth. Both cell lines began to exhibit apoptotic behavior 3–5 days after the removal of their corresponding growth factors from the medium (Figure 1B). The Ba/F3 cells showed relatively cleaner backgrounds after two weeks of growth factor deprivation, when 23% of EML cells were still alive (Figure 1C). The survival of EML in the absence of SCF could be attributed to non-specific response to other factors in the medium, such as the components in fetal bovine serum (FBS), or longer-lasting mitogenic signals. Therefore, Ba/F3 was selected as the basic cell line for BRADS. 

Initially, we attempted to incorporate a SIS-inducible element (SIE) containing a promoter responsive to STAT3 phosphorylation, which is the downstream signal of leptin receptor activation. Under the control of the SIE-promoter, membrane-tethered IL-3 was expressed to maintain the survival and proliferation of Ba/F3 in an autocrine manner (Figure 1A, bottom, dashed arrow). However, due to the weak response of SIE to IL-3 signal, the positive feedback of this circuit made the system unstable, sometimes causing self-activation in the absence of leptin signals. Hence, we adopted the ectopic stable expression of human leptin receptor (*h*LeptR) on Ba/F3 cells using the Piggy-Bac transposon system, which directly responds to leptin receptor activation (Figure 1A, bottom, hollow arrow). To validate this BRADS system, leptin was added to the medium in soluble form or constructed in cells in the membrane-tethered manner (autocrine), where it was N-terminally fused with the IL-2 signal peptide and C-terminally fused with a linker and the transmembrane domain of PDGFR [10] (Figure 1A). The autocrine form makes each cell an individual screening unit and enhances the signal upon surface expression, regardless of the concentration and duration of soluble-form cytokine exposure. The results showed that both soluble and membrane-tethered leptin maintained the survival and proliferation of Ba/F3-*h*LeptR well (Figure 1D). The effect of membrane-tethered leptin was even higher than that of 1 μM of soluble leptin. Single cell clones of the Ba/F3-*h*LeptR validated by leptin-induced proliferation were selected as the BRADS cell strains and preserved for further screening of human leptin receptor agonists.

### 3.2. Screening Based on the BRADS System

Considering that the capacity of BRADS screening is limited by the number of cells, a focused scFv library with a diversity of at least 10^6^ derived from a naïve library by one-step panning against the extracellular domain (ECD) of human leptin receptor was used [22]. This reduced the non-specific activation mediated by other cytokine receptors and reduced the capacity suitable for the BRADS system. The screening procedure is briefly illustrated as Figure 2A. We packaged this focused library in lentivirus in the autocrine form described above and used it for transduction to the BRADS cells. To increase the throughput of the screening, spinfection plus polybrene was used to enhance the transduction efficiency of the concentrated lentiviral antibody library stock, achieving a transduction rate as high as 60% (Figure 2B). The focused library was also transduced using a retrovirus system with a similar transduction rate to that of the lentivirus system (Figure 2B). To ensure sufficient coverage of the focused library with ~10^6^, a total of 50-fold excessive cells (5 × 10^7^ cells) were used, taking into account the transduction rate. 

The initial factor IL-3 was withdrawn from the medium the day after spinfection, when the antibodies started to be expressed (Figure 2C). The culture medium without IL-3 was refreshed every 3–4 days. No antibiotics or selective metabolic pressure was needed throughout the process. After two weeks of culture, the control BRADS cells without library transduction ceased proliferation, with most of them undergoing apoptosis, while the cells transduced with the focused library continued to proliferate (Figure 2D,E). A longer culture period for screening may further reduce the background, but it may also cause the loss of positive hits due to differences in proliferation rate among cell clones. A total of four independent screenings were performed, with one batch failing to enrich any surviving cells. The outputs of the remaining three screenings were pooled together to avoid batch biases.

The proliferating cells were collected, and their genomic DNA was purified. A primer pair flanking the scFv sequence was employed to amplify the enriched sequences, followed by Sanger sequencing (80 sequences). A total of four distinct clones were identified (B12, E06, B01, and C03, Figure 2F). Among them, clone B12 exhibited the highest enrichment, with 46 repeats, while clone E06 has 4 repeats, and the other two clones have only single hits. Interestingly, the most enriched clone, B12, with 46 repeats, has the same heavy chain variable domain (VH) sequence as the one previously selected by phenotypic screening with β-lactamase reporter from the same antibody library [22], Ab06, indicating the robust stability of the BRADS system (Figure 2F). We then aligned the two clones with repeats (B12 and E06) with the two clones identified in the previous screening using β-lactamase reporter (Ab06 and Ab11) and revealed a conserved FDY motif at the end of the CDRH3 region (Figure 2G). Further germline analysis showed a preference for the IGHJ gene 4*02, where the FDY motif resides (Figure 2G). None of these four clones has somatic hypermutation in V_H_. This result demonstrates the power of a combinatorial antibody library with a high diversity to circumvent the immune tolerance toward human endogenous protein targets.

### 3.3. Characterizations of B12 Clone

We constructed the B12 clone in scFv-hFc format in a pFUSE vector and expressed its antibody in 293F cells. The binding feature of the purified soluble B12 scFv-hFc antibody (sB12) to leptin receptors on live cells was characterized first. By flow cytometry, we titrated the apparent binding affinity of sB12 to leptin receptor-expressing Ba/F3 cells with an EC_50_ of 1.7 ± 0.5 nM (Figure 3A). Almost all of the *h*LeptR-expressing cells showed binding with sB12 above 10 nM. As opposed to the control cells in the screening, the proliferating cells strongly indicate the agonist activity of the enriched antibodies. Therefore, we tested and confirmed the activity of sB12 to stimulate the proliferation of the engineered BRADS Ba/F3 cells with an EC_50_ of around 1.4 nM (Figure 3B). sB12 was also validated to activate the leptin receptor using an SIE-GFP reporter in 293T cells to approximately 30% at a concentration of 3.3 nM (500 ng/mL), while leptin at the same concentration activated over 50% of the cells (Figure 3C). In another reporter system, the agonist activity was determined by an SIE-luciferase reporter with an EC_50_ of 2.2 nM, (Figure 3D). Unlike human leptin, the EC_50_ values of sB12 are approximately three orders of magnitude higher than that of human leptin in the two SIE reporter systems. Moreover, sB12 appeared to be a partial agonist of *h*LeptR, with approximately 70% of the full efficacy, as shown in both the SIE-GFP and SIE-Luciferase reporter systems (Figure 3B,D). In addition to the reporter assays, we further characterized the endogenous signaling induced by leptin and the surrogate antibody sB12 (Figure 3E). We checked the phosphorylation levels of STAT3 and STAT5 and found that, similar to the human leptin, sB12 performed comparably in activating STAT3 phosphorylation and highly activating STAT5 phosphorylation in the engineered BRADS cells.

## 4. Discussion

Phenotypic screening represents a milestone in pharmaceutical technologies. However, owing to its several unsolved deficiencies, such as the relatively high background signals, low throughput, the need for specific equipment, and the high cost of assay materials, its full potential in translation remains untapped. Addressing these issues could greatly enhance its application scope and boost drug discovery. 

In this study, we introduce a novel phenotypic screening approach that leverages cell survival as a selection criterion for agonists of cytokine receptors, taking the human leptin receptor as a proof-of-concept. Control cells lacking agonist stimulation almost ceased proliferation several days after withdrawal, with most becoming apoptotic within two weeks. In contrast, the clones with agonist activity made the relevant cells expand several fold. Therefore, this survival-based screening achieved the self-enrichment of positive clones, amplifying the desired signals throughout the screening process and ultimately minimizing background noise, as non-specific proliferation due to exogenous factors is random and transient. Moreover, the inherent self-enrichment of this method eliminates the need for reagents and high-throughput equipment for readouts or sorting, meaning costly labeling substrates, antibodies, array analysis by plate reader, cell imaging, and flow cytometry can be avoided. Another advantage of this survival-based screening is that it boosts high-throughput capabilities. To coordinate this survival-based screening approach, we adopted an autocrine system for the phenotypic library screening [10], in which most cells were transduced with a couple of antibody clones when we controlled the multiplicity of infection (MOI) and made the transduction rate close to 60% [24]. Each transduced cell is an individual screen unit, allowing for pooled library screening. Therefore, this method’s throughput is primarily constrained by the cell number, which can easily exceed 10^7^ in a single flask, particularly in suspension cultures. 

For this study, a naïve antibody library was pre-screened against the ECD of the human leptin receptor, narrowing down the diversity to match the screen capacity and focusing on the receptor-binding clones. This capacity size is applicable to most cytokine libraries. In this study, despite conducting four independent screenings, one batch revealed no enriched proliferating cells in the end, indicating the need for optimization in library construction and cell transduction efficiency to mitigate hit loss. However, the specificity of the positive screenings could be confirmed by using this batch as a control.

Addressing the issue of low sensitivity, or false negatives, is also critical, especially in cytokine receptor signaling, where a concentration threshold is necessary for receptor activation [10]. At concentrations below the threshold, a true agonist may still fail to activate some cells within the receptor. In pooled screening, this false negative can result in the loss of rare hits in huge libraries, leading to selection failure. In the autocrine format used in this study, antibodies, once expressed, are anchored on the cell surface, leading to sustained activation of the receptor. This setup mimics the concentration of soluble factors in close proximity to the receptor, thereby enhancing the signal without amplifying background noise [14,25].

Factor-dependent cell lines like Ba/F3, EML, and FDCP have been commonly used in studies of cell proliferation, survival, self-renewal, and so on [26,27,28,29,30,31]. Yet, to our knowledge, there has not been a report of successful high-throughput screening for agonists from a library with a diversity above millions. This gap can be attributed to the inherent limitations of arrayed screening based on cell phenotypes, including the low throughput, low gene delivery efficiency to live cells, and the challenge of adaptions to receptors with diverse characteristics. In this study, we implemented a pooled screening in autocrine format, bypassing the limitations of arrayed screening, accommodating huge libraries. We utilized a lentiviral vector for library delivery and optimized the transduction efficiency to around 60% in the Ba/F3 BRADS strain. This optimization fulfills the requirements of the BRADS screening system. The whole BRADS screening process is carried out under standard mammalian cell culture conditions without the necessity for specialized equipment or reagents. Thus, we present a high-throughput, low-background, and cost-effective screening system. 

In the case shown in this study, we identified an agonist antibody for the human leptin receptor. The leptin receptor belongs to the class I cytokine receptor family, sharing structural homology with many hematopoietic factor receptors [32]. It signals via the JAK/STAT pathways, promoting cell proliferation [32]. For such receptor tyrosine kinases (cytokine receptors), our method requires no further engineering beyond the ectopic expression of the receptor of interest. However, the applicability of BRADS extends beyond receptor tyrosine kinases. For example, to select artificial soluble agonists for Notch receptors, one could engineer the SynNotch system, wherein the reporter gene is replaced with the mIL-3 gene [33]. Consequently, Notch agonists in the library would induce the expression of mIL-3, stimulating BRADS cell proliferation (Figure 1C, dashed arrow). More complex circuits allow for the expansion of BRADS applications to nearly any kind of biological receptor, even intracellular targets. Additionally, BRADS is not confined to biologic combinatorial libraries. Pooled chemicals libraries such as DNA-encoded libraries are also compatible with this system [34].

In conclusion, to overcome the existing problems in phenotypic screening, we have presented a novel high-throughput phenotypic screening method for agonists of biological receptors on live cells based on cell survival. The method is characterized by its low background and high sensitivity and named BRADS. We have demonstrated its efficacy by describing the selection of a human leptin receptor agonist antibody, B12, from a combinatorial antibody library with huge capacity, showcasing its potential to identify rare hits effectively.

## Figures and Tables

**Figure 1 biomolecules-14-00457-f001:**
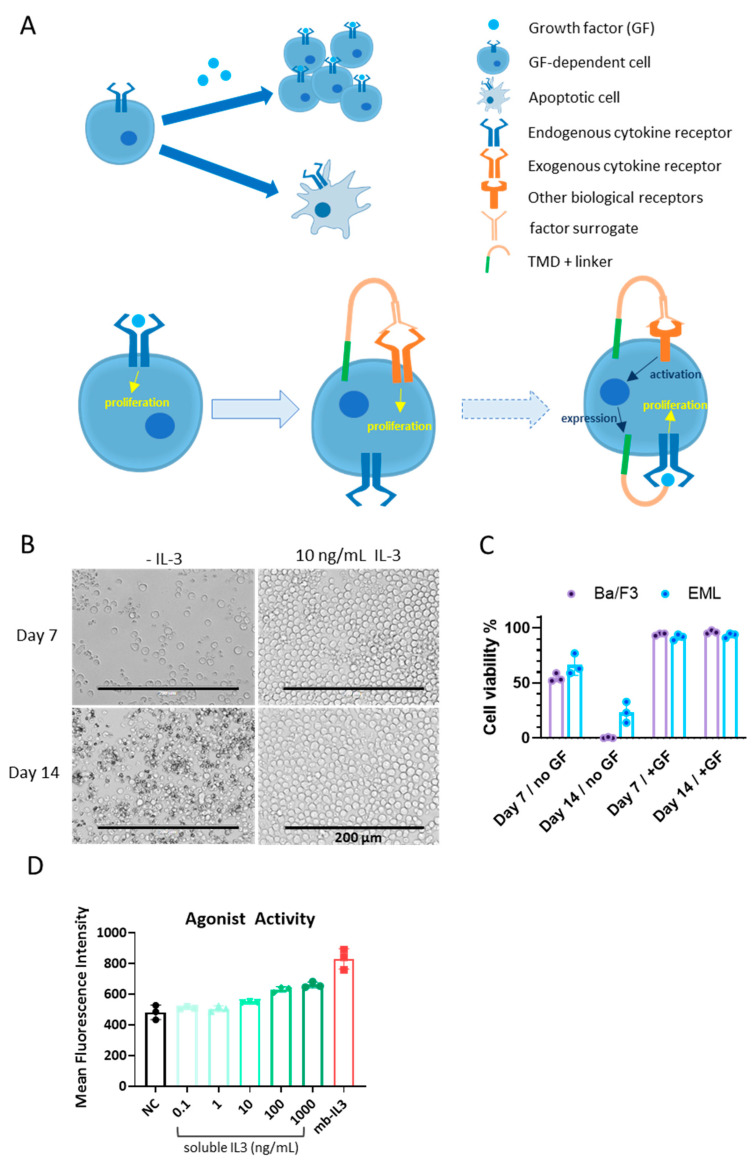
Establishment of the BRADS cells. (**A**) A brief illustration of the BRADS with the figure legend items showed beside. GF-dependent cells proliferate in the presence of the corresponding GF but become apoptotic in the absence of the GF (upper left). The GF-dependent cell could be generated into a BRADS cell when an exogenous cytokine receptor of interest is constructed on the cell, which can be activated by the surrogate in the membrane-bound autocrine format (bottom, hollow arrow). The cytokine receptor (receptor tyrosine kinase) could be replaced by other kinds of biological receptors when the exogenous membrane-bound GF is constructed as the reporter gene of the receptor in the cell (bottom, dashed arrow). (**B**) The morphology of Ba/F3 cells cultured in the absence or presence of 10 ng/mL murine IL-3 for 7 days and 14 days. (**C**) The percentage of live cells cultured in the absence or presence of growth factors (GFs) for 7 days and 14 days. We used 10 ng/mL murine IL-3 as the GF for the Ba/F3 cell line (purple columns), and 200 ng/mL murine SCF was used as the GF for the EML cell line (blue columns). (**D**) The agonist activity of IL-3 in soluble form and membrane-bound form (mb-IL3) was determined as the mean fluorescence intensity of GFP in the SIE-GFP Ba/F3 cell detected by flow cytometry.

**Figure 2 biomolecules-14-00457-f002:**
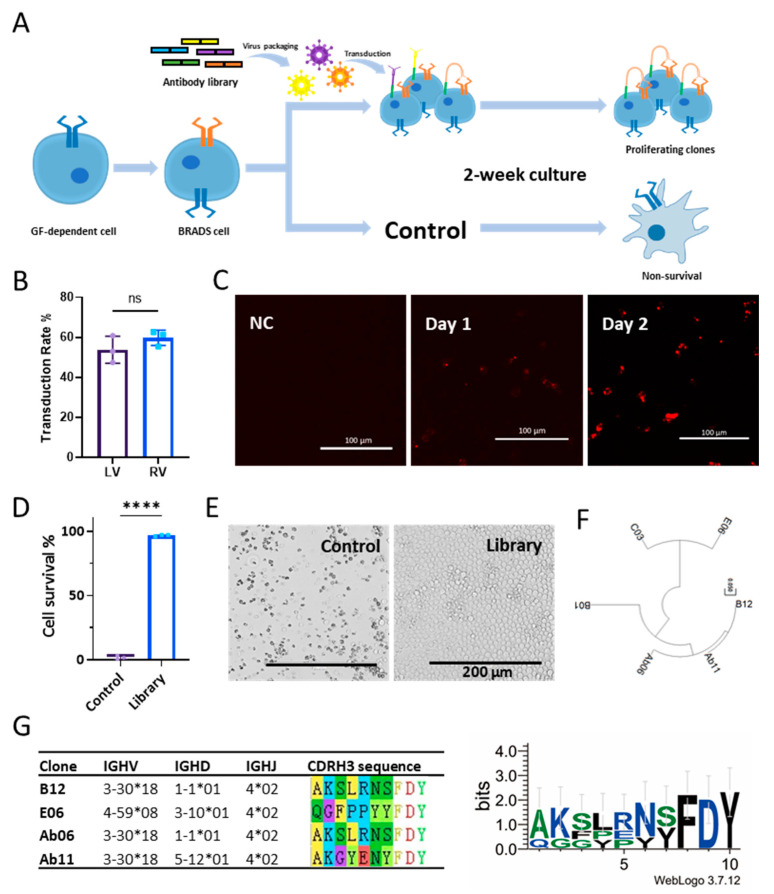
Agonist screening for human leptin receptor based on BRADS. (**A**) Illustration of the screening process. The *h*LeptR-focused antibody library (colored sticks) was cloned into the lentivirus vector in autocrine format, followed by lentivirus packaging (colored spiked circles) and transduction into the *h*LeptR BRADS cells. The cell pool was cultured for 2 weeks for the proliferation of positive clones and elimination of negative clones. In the end, the proliferating cells were harvested for the subsequent sequencing analysis. (**B**) No significant difference (ns, *p* > 0.05) in the virus transduction rate for the BRADS cells was found between the lentivirus (LV) and retrovirus (RV) in this study. (**C**) The membrane-bound antibody started to be expressed on the cell surface the day after library transduction. NC represents no transduction. (**D**) The percentages of viable cells of the library transduced cell sample and the control cell sample without transduction in the end of the screening process. (**E**) Representative images of cell morphology of the two groups in (**D**). (**F**) The four distinct antibody sequences (B01, B12, C03, and E06) identified by PCR and Sanger sequencing of the collected proliferating cells plus the two antibody sequences (Ab06 and Ab11) identified by phenotypic screening using β-lactamase reporter in our previous work [22] were aligned using MAGA software (version 11.0.13). (**G**) The germline sequence analysis of antibody heavy chain and CDR3 sequencing alignment of the two enriched clones with repeats in sequencing results (B12 and E06) and Ab06 and Ab11 are shown in the left table. The share of 4*02 IGHJ germline gene and an FDY motif is presented in the right plot (WebLogo). All bar plot data are presented as mean ± SD. A two tailed *t*-test was employed for the analysis of the significance of the differences between the two groups (**** *p* < 0.0001). Antibody germline gene number is structured as “family number(-gene segment type number)*allele number”.

**Figure 3 biomolecules-14-00457-f003:**
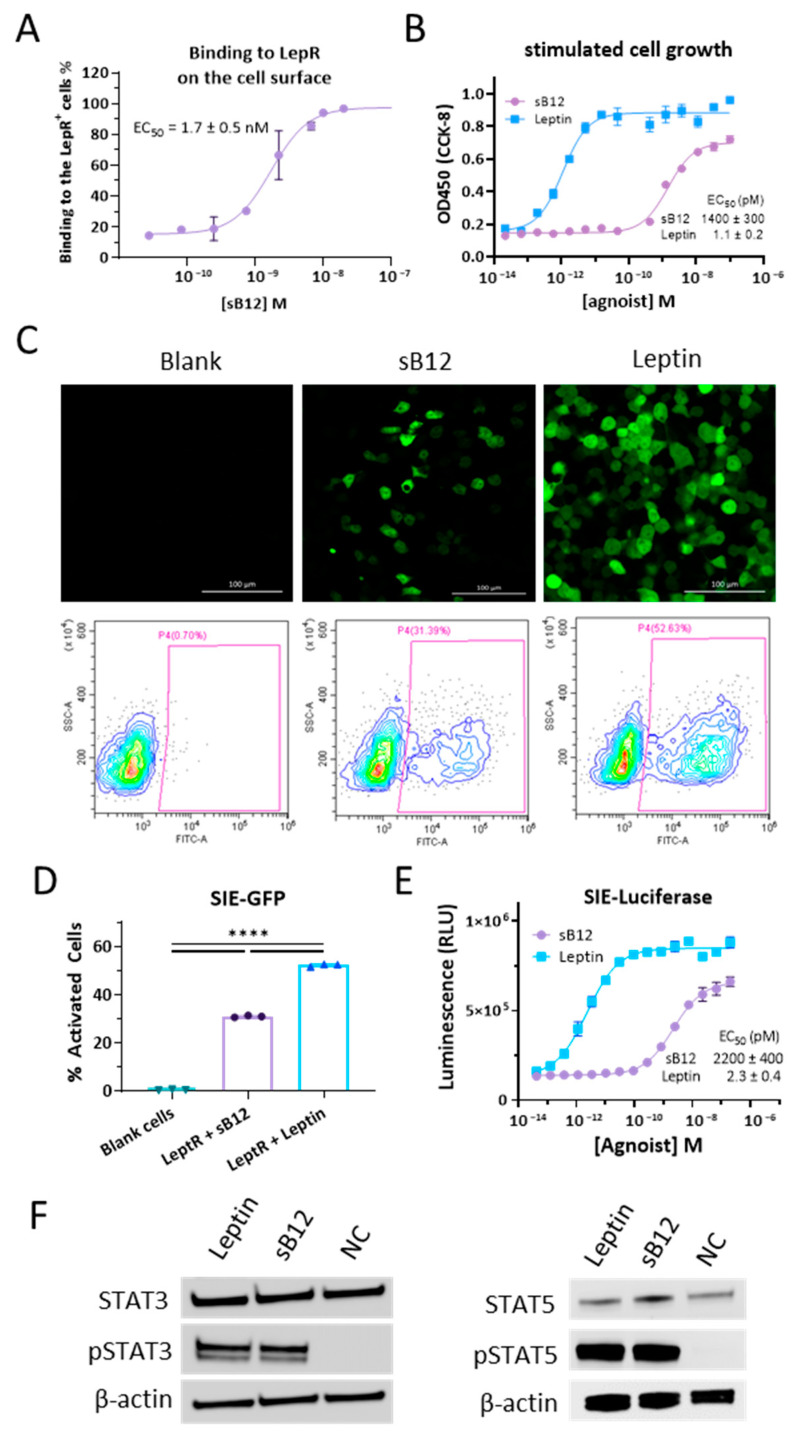
Characterization of B12 clones. (**A**) Binding affinity of sB12 to the *h*LeptR on cell surface, determined by flow cytometry. The dose-dependent effect curve was fitted in GraphPad with the EC_50_ value shown inside the panel. (**B**) Activation of proliferation of *h*LeptR BRADS cells by sB12 at the different concentrations indicated in the Materials and Methods section. The effect curve was fitted in GraphPad with the EC_50_ value shown inside the panel. (**C**) Representative fluorescence images and flow cytometry results for the SIE-GFP-292T reporter cells stimulated by 500 ng/mL sB12 and human leptin. “Blank” means SIE-GFP-292T cells without agonist stimulation. (**D**) Statistical analysis of SIE-GFP activation. “Blank cells” represents SIE-GFP-293T without stimulation. “NC” represents stimulation by an irrelevant antibody. Data are presented as mean ± SD. An unpaired ANOVA was employed for the analysis of the significance of the differences among the groups (**** *p* < 0.0001). (**E**) Activation of SIE-Luciferase-292T reporter cells by human leptin (blue) and sB12 (purple) at the different concentrations indicated in the Materials and Methods. The effect curve was fitted in GraphPad with the EC_50_ value shown inside the panel. (**F**) Western blotting results of STAT3 or STAT5 and phosphorylated STAT3 or STAT5 (pSTAT3, pSTAT5) level induced by 500 ng/mL human leptin or sB12. NC represents the control cell sample without agonist stimulation.

## Data Availability

Raw data are available from the corresponding authors upon reasonable request.

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
