# Peer review of "Selection of Leptin Surrogates by a General Phenotypic Screening Method for Receptor Agonists"

_biomolecules, 2024, doi:10.3390/biom14040457_

Round 1

Reviewer 1 Report

Comments and Suggestions for Authors

Overall the manuscript is well written. It describes the development and proof-of-concept testing of a phenotypic screening method using biological receptor activation-dependent cell survival (BRADS). The example of using the leptin receptor as an example was well planned and executed. It was very innovative to use a lentiviral antibody library that was previously developed. The data results are supported by the finding that the top clone identified was also found in a parallel reporter screen using the same antibody library for leptin receptor. Validation experiments were appropriate and convincing. Some minor controls could have made the screening results and proof of concept stronger, including a better negative control than the non-transduced cells.  Perhaps using an unrelated antibody pool (eg. a different receptor target) to show that the antibody clones were specific for the target receptor is a more fair control to compare. Beyond testing downstream effector proteins in a cell model it would be nice to show more evidence of specificity to the leptin receptor. What about an in vivo model to further validate the HTS hit?  Despite these minor points, the manuscript and methods are novel and could potentially be used for other receptor targets of interest and with different pools of effector agents.

Author Response

Response:

  1. We appreciate the reviewer for the insightful suggestions. Due to the large diversity of the original library (≥1011), we constructed a sub-library (≤107) for the BRADS by a single round of phage panning against the leptin receptor ECD. Despite for the focused sub-library, it is not always successful to get proliferating cells containing the positive hits in the end. Actually, we undertook 4 independent screenings for the leptin receptor via BRADS, with 3 of them yielding enriched proliferating cells. These cells were then pooled together for further analysis. The failure to obtain positive hits in one screening highlights the inherent limitations in library construction and the efficiency of transduction into cells. This screening, which did not result in proliferation, could serve an important role as a control, ruling out the unspecific effects of antibody expression and viral transduction on the cells. We have incorporated these clarifications and insights into our manuscript, in the Results (Line 239-241) and Discussion (L351-355), highlighted in yellow.
  2. We concur with the reviewer on the critical importance of in vivo validation of the hits. We are now validating the antibodies in several obesity and inflammatory models, along with the leptin surrogate hits derived from other screenings besides BRADS. Such endeavors are inherently time-consuming and do not focus on the BRADS so we suggest not including the results in this manuscript.

Reviewer 2 Report

Comments and Suggestions for Authors

Reviewing the manuscript entitled "Selection of leptin surrogates by a general phenotypic screening method for receptor agonists”, the authors, after two weeks of cell culture, were able to identify a human leptin surrogate without the need for specialist high-throughput tools or reagents. In cell validation experiments, this surrogate successfully mimics native human leptin function. The results demonstrate the efficacy of BRADS and point to its possible application to a wide class of biological receptors, including GPCRs and Notch.

In my opinion, the manuscript fills an important gap in high-throughput methodologies for the discovery of new active compounds. In my opinion, the manuscript could be published if some minor issues are addressed.

1.      In the abstract, the sentence "However, efficient screening methods for identifying rare hits within immense libraries remain elusive" should be replaced because many efficient screening methods for discovering hits are now known.

2.      The introduction lacks the corresponding bibliographies for each proposal. Please introduce them.

3.      The same applies to the discussion. Corresponding bibliographies should be included.

4.      Editing of the English language is needed in the text.

Comments on the Quality of English Language

Extensive editing of the English language is required.

Author Response

Response:

1. We have revised the sentence as suggested. As we mentioned in the Introduction, although several studies have addressed the issue (Ref. 8-13), there are still significant unmet needs in this field (Line 47-50).

2. We have supplemented some references in the revised manuscript now, including Ref. 3, 22, 23, 32 and 33.

3. Please see above.

4. We have edited the language throughout the manuscript (in the Track Changes mode).